# Efficient Representative Volume Element of a Matrix–Precipitate Microstructure—Application on AlSi10Mg Alloy

Chantal Bouffioux [1], Luc Papeleux [2], Mathieu Calvat [3], Hoang-Son Tran [1], Fan Chen [1], Jean-Philippe Ponthot [2,*], Laurent Duchêne [1] and Anne Marie Habraken [1,*]

1    Materials and Solid Mechanics, University of Liège, Allée de la Découverte 9 B52/3, B-4000 Liege, Belgium; chantal.bouffioux@uliege.be (C.B.); hstran@uliege.be (H.-S.T.); fchen@uliege.be (F.C.); l.duchene@uliege.be (L.D.)
2    Aerospace and Mechanical Engineering, University of Liège, Allée de la Découverte 13 A B52/3, B-4000 Liege, Belgium; l.papeleux@uliege.be
3    Materials Science and Engineering, University of Illinois, 201 Science and Engineering Building, 1304 W. Green St. MC 246, Urbana-Champaign, IL 61801, USA; mcalvat@illinois.edu
\*    Correspondence: jp.ponthot@uliege.be (J.-P.P.); anne.habraken@uliege.be (A.M.H.)

**Abstract:** In finite element models (FEMs), two- or three-dimensional Representative Volume Elements (RVEs) based on a statistical distribution of particles in a matrix can predict mechanical material properties. This article studies an alternative to 3D RVEs with a 2.5D RVE approach defined by a one-plane layer of 3D elements to model the material behavior. This 2.5D RVE relies on springs applied in the out-of-plane direction to constrain the two lateral deformations to be compatible, with the goal of achieving the isotropy of the studied material. The method is experimentally validated by the prediction of the tensile stress–strain curve of a bi-phasic microstructure of the AlSi10Mg alloy. Produced by additive manufacturing, the sample material becomes isotropic after friction stir processing post treatment. If a classical plane strain 2D RVE simulation is clearly too stiff compared to the experiment, the predictions of the stress–strain curves based on 2.5D RVE, 2D RVE with no transversal constraint (called 2D free RVE), and 3D RVE simulations are close to the experiments. The local stress fields within a 2.5D RVE present an interesting similarity with 3D RVE local fields, but differences with the 2D free RVE local results. Since a 2.5D RVE simplifies one spatial dimension, the simulations with this model are faster than the 3D RVE (factor 2580 in CPU or taking into account an optimal parallel computation, a factor 417 in real time). Such a discrepancy can affect the FEM$^2$ multi-scale simulations or the time required to train a neural network, enhancing the interest in a 2.5D RVE model.

**Keywords:** Representative Volume Element; 2.5D numerical model; AlSi10Mg; additive manufacturing; microstructure; hardening behavior

## 1. Introduction

The 2D or 3D finite element simulations of Representative Volume Elements (RVEs) are increasingly popular as they help the researchers to understand the strength mechanisms within the loaded materials. A careful design of the RVE is crucial to ensure an accurate and reliable representation of material microstructures, capable of effectively predicting macro-scale properties, as demonstrated by significant validation with experimental data.

As pointed in reference [1], the RVE methodology encompasses two primary approaches: statistically modeling the microstructure as a virtual representative volume [2], or discretizing a statically representative real volume [3]. Based on microstructure features such as spherical particles, randomly distributed needle-like precipitates in a matrix, isotropic polycrystalline material, or strong anisotropy due to texture or grain shape, the choice between 2D and 3D RVEs is required to investigate the mechanical response under various loading conditions. The main focus of this work is to explore an alternative to 3D

RVEs, also called 2.5D RVEs, in order to save the CPU time, particularly in the context of isotropic materials featuring a soft matrix containing hard particles. The 2.5D RVE approach is defined by a one-plane layer of 3D elements and relies on specific constraints in the out-of-plane direction to model any isotropic material presenting a matrix with particles. Subsequently, this introduction discusses the range of simulation goals achievable with RVEs, exemplified through several metal-based examples.

The quality of RVE results explains their use in virtual material and process design, as RVEs can predict material properties based on a given microstructure. They are now integrated into materials engineering to enhance material design. For instance, Maity et al. [3] use 2D RVEs to understand the effect of the addition of Mn in the Al-12.6 Si alloy on bulk hardness, yield stress, and fracture propagation. Indeed, Mn presence varies the microstructure morphology and the micromechanical response of the alloy. Using 3D RVEs, Shalimov and Tashkinov [4] demonstrate that for cell porous gold crystals, the random morphology can play an as important role in the mechanical tensile curve as the pore fraction. Akbari et al. [5] study the link between the yield surface of a polycrystalline brittle material and its microstructure by using 2D RVEs, including cohesive elements to model the grain interface behavior. Sun and Jain [6] simulate the AA7075 elastic behavior based on its complex microstructure involving irregularly shaped $Al_3Fe$ particles, elliptical $MgZn_2$ ones, and needle-like $CuAl_2$ ones. Their 3D RVE models accurately estimate the effective elastic properties of the AA7075-O sheet. Reis et al. [7] use 2D RVEs to study heterogeneous ductile materials, employing non-local formulations to predict the damage localization path. As pointed out by Gillner and Münstermann [8] while working on 2D RVEs of a ferrite pearlite steel, the RVE results can decrease the cost of experimental campaigns by identifying the target microstructure able to increase fatigue lifetime. The above examples confirm the use and the interest in both 2D and 3D RVEs. Based on statistical data analysis, a systematic comparison of local results has been carried out by Qayyum et al. [9] to conclude that the 3D RVEs provide better quantitative results than experiments, whereas the 2D RVEs already provide appropriate qualitative information about the damage initiation sites as well as an accurate macroscopic stress–strain response for a small and medium plastic range, which is often sufficient for many practical applications. In a pragmatic way, based on [10] and their own previous study [11], Qayyum et al. [9] suggest to generalize that damage initiation in 2D RVEs occurs for a global strain of 6% earlier than the damage initiation in 3D RVEs for Dual-Phase (DP) steel. Indeed, 2D RVE models can definitively be exploited to understand hardening or damage mechanisms, as shown in [12], as well as study the effect of a ferrite–martensite interface.

In materials science, the machine learning approach is currently applied for material design. See, for instance, the DP steel design through 3D RVE simulations [13], where an accurate use of 2D RVEs could significantly reduce computational costs, which is crucial when generating the large datasets needed to train machine learning models. This efficiency comes from the simpler geometry and numerical requirements of 2D simulations compared to 3D ones, allowing for a quicker exploration of a wider range of material behaviors and parameters. Furthermore, 2D RVEs make it easier to visualize and interpret the material microstructure and its properties, which is beneficial during the early stages of a material design (process parameter optimization as well as post-processing operations). These observations explain our efforts to quantify the accuracy of a 2D RVE versus a 3D one for matrix–precipitate material and to develop a specific 2.5D RVE.

Laser Powder Bed Fusion (L-PBF) produces materials with high strength [14,15], whereas Friction Stir Processing (FSP) reduces the out-of-equilibrium microstructural state without causing excessive softening [16]. A new FSP microstructure has the added benefit of closing porosities, which positively impacts ductility and fatigue behavior within the single FSP pass post-processed material [17]. In the present article, an L-PBF material post-processed by FSP was chosen as the case study. The as-built L-PBF AlSi10Mg material investigated has undergone testing in two orthogonal tensile directions [18] and has been extensively characterized in previous research [14,18]. Its microstructure is defined by an

interconnected Si network forming cells of different sizes according to their position within each melt pool zone. Post-processed by FSP [15,17], the Si network is globularized into Si particles. The alloy forms an aluminum matrix called the $\alpha$ phase ($\alpha$-Al) containing Si-rich precipitates which exhibit macroscopic isotropic behavior. As pointed out by different authors [19,20], combining FSP with additive manufacturing (AM) on an industrial scale presents both opportunities and challenges. FSP can significantly improve the mechanical properties of AM parts by refining microstructures and addressing defects like porosity and anisotropy, making it valuable for industries, such as aerospace and automotive, where high material performance is crucial. However, large-scale implementation still requires overcoming practical challenges, particularly the integration of FSP into highly automated AM workflows. This would involve developing efficient, high-throughput systems to keep up with production rates, and addressing issues like tool wear and process control, which can be influenced by part geometries. Recent research has been increasingly focused on combining AM and FSP to leverage the strengths of both processes [19–21].

Hereafter, we investigate the ability of 2D and 3D RVE simulations to accurately predict the measured tensile hardening curve of a typical Al alloy microstructure, formed by L-PBF and post-processed by FSP. In our simulations, currently the absence of cohesive elements limits the detailed analysis of fracture mechanisms, a topic we intend to address in future research as well as the fatigue behavior. As demonstrated by Yuan et al. [22], who validated their plane strain RVE results with a tensile test before addressing fatigue prediction, the accuracy of 2D and 3D RVEs is hereafter checked for the macroscopic tensile stress–strain curve.

To define an RVE for this L-PBF FSP AlSi10Mg material, it is necessary to determine its size, its finite element mesh, and the applied boundary conditions [23]. These choices are not independent as periodic boundary conditions allow reducing the size of the RVE volume. The selected mesh generator has to handle both geometric and boundary condition periodicity [24,25]. Generating RVE meshes with periodic conditions and flexible refinement for the matrix and particles is a complex task. Specific tools like Neper [26], Digimat [27], and GMSH v 4.11.0 [28] have been investigated for this purpose. Hereafter, it has been decided to develop scripts based on GMSH v 4.11.0 to build the desired microstructure mesh, since GMSH v 4.11.0 is quite flexible and comes with a well-documented Python 3.8 interface. This capability enables the creation of intricate scripts, including algorithms that can reconstruct meshes using only quadrangle elements [29,30]. Additionally, features such as imposing periodicity conditions on the mesh and handling complex geometric operations such as cuts, fusions, and intersections between geometrical entities make it straightforward to use with support from the OpenCascade library [31]. In the ULiege Lagamine FEM software [32], to keep accuracy, it was chosen to mesh the 2D cut of the L-PBF FSP AlSi10Mg microstructure using only quadrangle elements to avoid any locking.

Figure 1 summarizes the procedure described in this article to build a 2.5D RVE: a one-plane layer of 3D finite elements relying on specific constraints in the out-of-plane direction. This approach is able to model any isotropic material presenting a matrix with particles. It keeps the low computation time of a 2D RVE compared with a 3D RVE.

The structure of this article is as follows: Section 2 presents the microstructure data collected for designing the RVEs, along with the experimental tensile curve. Section 3 summarizes the various 2.5D RVE models tested under tension, including their mesh size, specific boundary conditions tailored to mimic a 3D behavior, and the material constitutive laws employed. Section 4 analyzes the predictions obtained from the 2.5D RVE simulations versus experimental results, highlighting their superiority over simpler 2D cases such as a simple membrane in a plane strain for instance. Section 5 compares all the 2D and 3D RVE results with experimental data, while conclusions and future perspectives are discussed in Section 6.

| 2.5D RVE for an isotropic material characterized by a matrix and particle microstructure |
|---|
| 1.   Perform physical tensile tests (check isotropy, generate a reference experimental stress–strain curve) |
| 2.   Characterize the microstructure: microscope image analyses (particle statistical features: sizes, shapes, orientations), if possible perform nanoindentation (behavior of matrix and particles) |
| 3.   Design a few 2D RVEs: different RVE sizes and particle distributions based on Step 2 |
| 4.   Mesh 2D RVEs with a geometric periodicity of particles within a plane, by 1 layer of 3D FEM elements |
| 5.   Generate 2.5D RVEs: application of periodic boundaries in the 2D RVE plane and define out-of-plane spring stiffness (1 stiffness for matrix springs and 1 for the particle springs) |
| 6.   Identify unknown material data and out-of-plane stiffness of matrix springs and particles by inverse analysis (the experimental targets are the tensile stress–strain curve of step 1 and isotropy) for 2.5D RVEs |
| 7.   Select "the 2.5D FEM RVE", best compromise between accuracy and computation time |

**Figure 1.** Flowchart of the procedure to build a 2.5D RVE.

## 2. Material Description

The material under investigation is an AlSi10Mg alloy manufactured by L-PBF and post-processed with a single pass of FSP. The process parameters employed to fabricate the samples in this study are detailed in Dedry et al. [33]. Figure 2a shows a typical microstructure observed by SEM [15] while its post-processing by "ImageJ 1.52a" allows a clear observation of the microstructure composed of globularized Si-rich particles within an α-Al solid solution (see Figure 2b). The studied zone of $11.4 \times 7.7 = 87.78$ μm$^2$ contains 157 Si hard particles embedded in the α-Al matrix. The smallest Si particles, i.e., area less than 0.011 μm$^2$ (Figure 2b), were not considered in this amount, as they do not affect the mechanical behavior of the RVE. This conclusion was derived from the sensitivity analysis of FE simulations [34] and from more than 100 experimental nanoindentations [33].

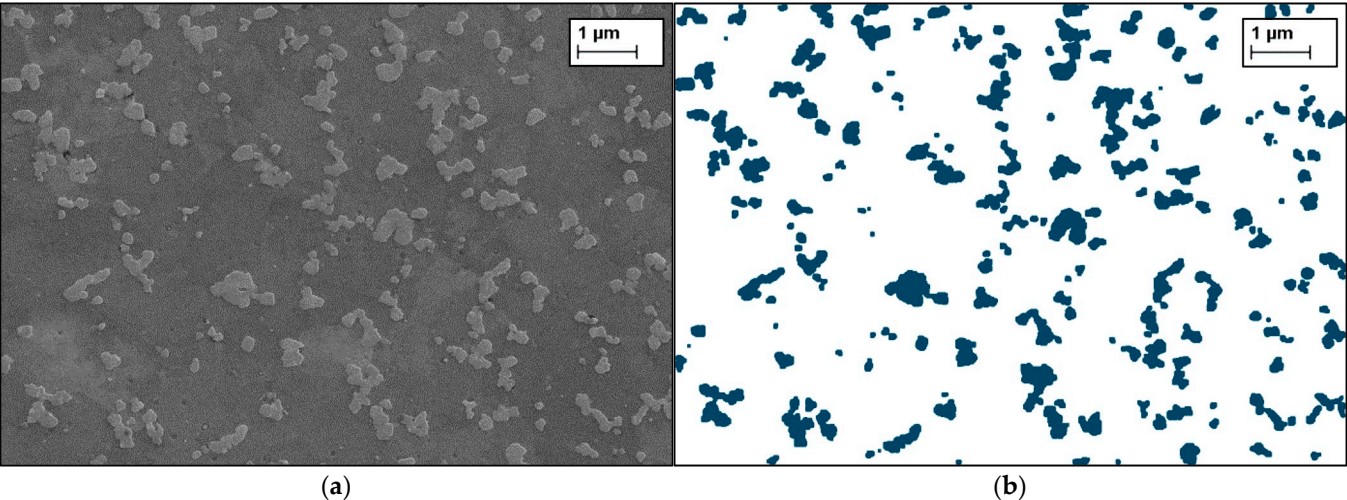

(**a**)                                                        (**b**)

**Figure 2.** (**a**) Microstructure of AlSi10Mg after FSP (Reprinted with permission from [15]. Copyright Year ELSEVIER), (**b**) particles of Si from "IMAGEJ 1.52a" analysis.

Each particle is described by several parameters: the equivalent diameter of a circle with the same area $\varnothing_{eq}$, the equivalent ellipse with its aspect ratio *AR* defined by Equation (1) varying between 0.247 and 0.941, and the particle orientation defined by the angle $\alpha$ between the horizontal axis of the image and the major axis of the ellipse. The *AR* is given as follows:

$$AR = b/a \tag{1}$$

where $2b$ and $2a$ are the lengths of the minor and major axes of the equivalent ellipse. The distribution of the particles with respect to their $\varnothing_{eq}$ is in the range of [0.118; 0.695] μm with a higher rate of small particles. The *AR* analysis shows few highly elongated equivalent ellipses (*AR* < 0.4). The angles $\alpha$ show no strong dominant direction. A detailed description

is available in the Supplementary Material. The well-known Pearson correlation coefficient *r* used to determine the degree of linear correlation between two variables is recalled in Equation (2):

$$r_{m,n} = \frac{cov(m,n)}{\sigma_m.\sigma_n} \tag{2}$$

If *m* and *n* are the variables, *cov(m,n)* is their covariance, and $\sigma_m$ and $\sigma_n$ are their standard deviations, respectively. No correlation exists when $r_{m,n} = 0$, while a perfect negative or positive correlation is found when $r_{m,n} = -1$ or $+1$, respectively. Here, the *r* coefficients (Table 1), computed for each pair of geometric parameters describing the Si particles, indicate a moderate (almost weak) correlation between *AR* and $\varnothing_{eq}$ (Figure 3a) and a weak correlation between both $\alpha$ and $\varnothing_{eq}$ (Figure 3b) and $\alpha$ and *AR* (Figure 4a). A corrected Pearson coefficient value is provided for the angle correlation, due to the periodic context. These microstructural statistical observations are used to generate the RVE models. If the blue dots in Figures 3 and 4 illustrate the low relationships between the particle size, shape, and orientation observed in the SEM image, the black bullets correspond to the 10 particles of the medium-B RVE model (see Section 3.1).

**Table 1.** Correlation factors r between the geometric parameters of the Si particles.

| AR vs. $\varnothing_{eq}$ | $\alpha$ vs. $\varnothing_{eq}$ | $\alpha$ vs. AR |
|---|---|---|
| −0.336 | 0.042 | 0.066 |

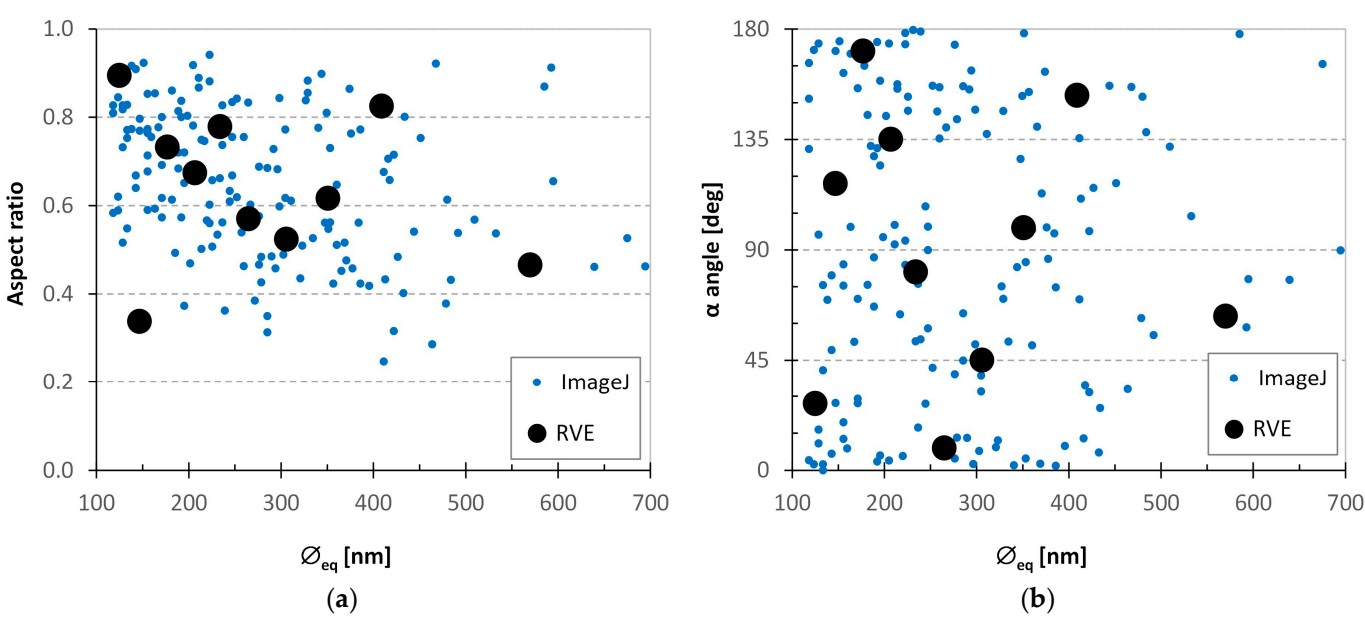

**Figure 3.** Si particle distributions in SEM images and in the "2D medium-B RVE model": (**a**) AR versus $\varnothing_{eq}$, (**b**) $\alpha$ angle versus $\varnothing_{eq}$.

In Figure 4b, the tensile test results of specimens, cut perpendicularly to the L-PBF building direction and parallel to the FSP tool displacement direction, are depicted (see process and test details in Zhao et al. [15] and Santos Macias et al. [17]). Very low experimental result scattering is observed, and the average curve will be henceforth used as a reference for the macroscopic material tensile behavior.

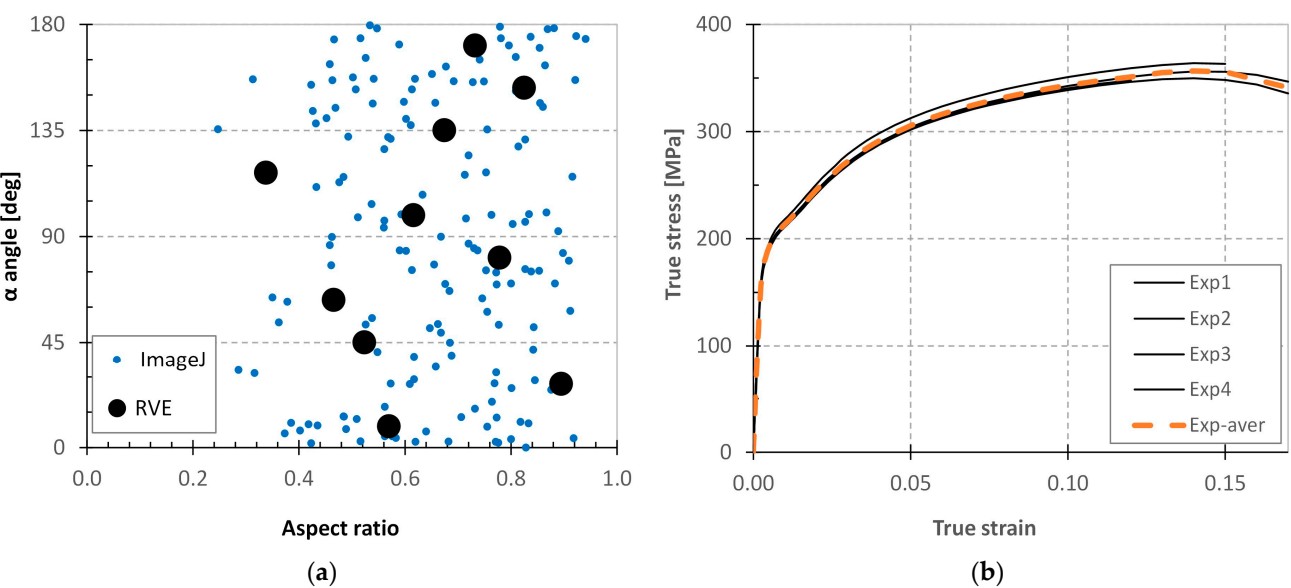

**Figure 4.** (**a**) Si particles distribution in SEM images and in the "2D medium-B RVE model": α angle versus AR, (**b**) tensile tests on AlSi10Mg (Exp i) and average curve (Exp-aver).

## 3. Numerical Model

Lagamine FEM software [32], developed in ULiege since 1985, was used to design the 2.5D RVE. This developed RVE was defined as a flat volume (a square cross-section and a low thickness) composed of a single layer of 3D elements with constraints on nodal out-of-plane displacements. If only classical J2 plasticity was exploited hereafter, note that the Lagamine FEM code can handle a crystal plasticity approach. See, for instance, Yuan et al. [35], where strain gradient crystal plasticity is used to examine size effects in Nickel samples. Hereafter, the 3D RVE models were implemented in the METAFOR code v3494 [36], another ULiege home-made software more focused on large simulations.

### 3.1. RVE Definition

To determine the ideal RVE size, three 2.5D square RVE sizes (called small, medium, and large) are studied and compared. They contain 5, 10, and 15 elliptical Si-rich particles and are characterized by 1.672, 2.365, and 2.896 μm sides, respectively. All these RVEs have an average surface of 0.559 μm$^2$ by particle, as in the experimental microstructure. Figure 5 shows the small and large RVE models while Figure 6 provides 2 distributions of particles in the medium RVE, called medium-A and medium-B. The sizes, shapes, and orientations of the Si particles are distributed in a comparable way to those of the SEM microstructure image (see the black bullet points in Figure 3a,b and Figure 4a for the medium-B model).

The Si particle positions are defined randomly. Table S1 of the Supplementary Material describes the 10 particles in the medium-B model. To avoid any bias in the periodic geometrical representation, the particles can cross the edges of the RVE. To ensure the periodicity, the missing particle parts are replicated on the opposite sides (Figures 5 and 6). Different meshes have been generated by a dedicated Python 3.8 script using the GMSH v 4.11.0 software API. The mesh density can be defined in the matrix and for each particle, allowing to refine the mesh where it is necessary. Each 2D RVE mesh is created with linear quadrangle elements, representing the α-Al matrix and the Si particles defined as ellipses. The periodic boundary conditions are directly applied on mirror nodes. The domain is extruded to generate one layer of 3D hexahedron elements defining the 2.5D model.

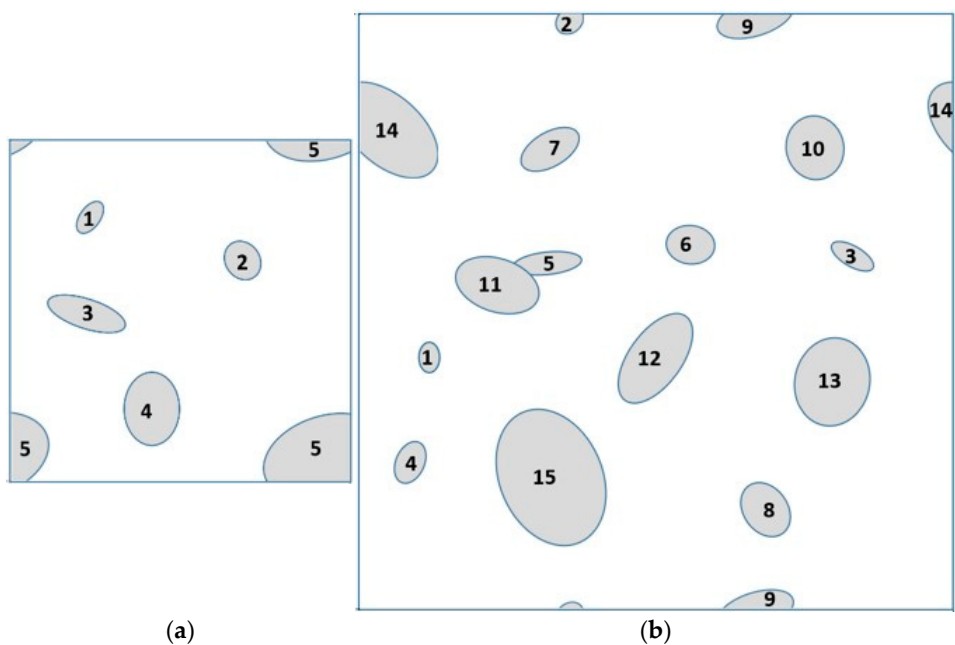

**Figure 5.** (**a**) Small RVE model of 1.672 μm side with 5 particles and (**b**) large one of 2.896 μm side with 15 particles of Si.

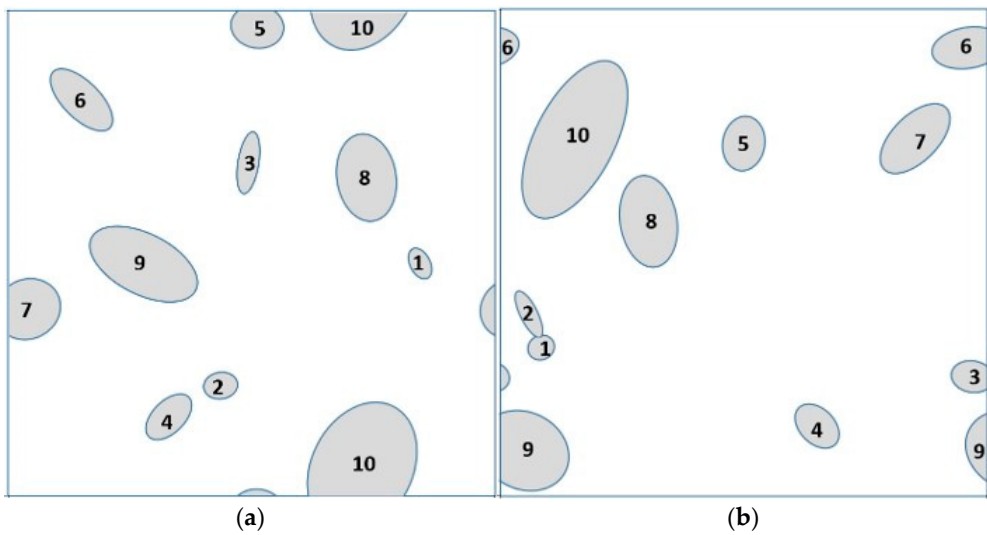

**Figure 6.** Medium RVE models of 2.365 μm side with two distributions of the 10 particles: (**a**) medium-A, (**b**) medium-B (each of the 10 particles are further described in Supplementary Materials).

Examples of coarse, intermediate, and refined meshes are plotted in Figure 7 for the medium-B case of 10 Si particles. The density of elements per μm$^2$ and the total number of elements in all the tested models are summarized in Table 2. Note that all the meshes have one layer of height-node brick BWD3D elements, implemented in the Lagamine code by Zhu and Cescotto [37]. This type of element is based on the non-linear three-field HU-WASHIZU variational principle of stress, strain, and displacement [38,39]. It uses a mixed formulation adapted to large strains and large displacements with a reduced integration scheme (only one integration point per element) and an hourglass control technique.

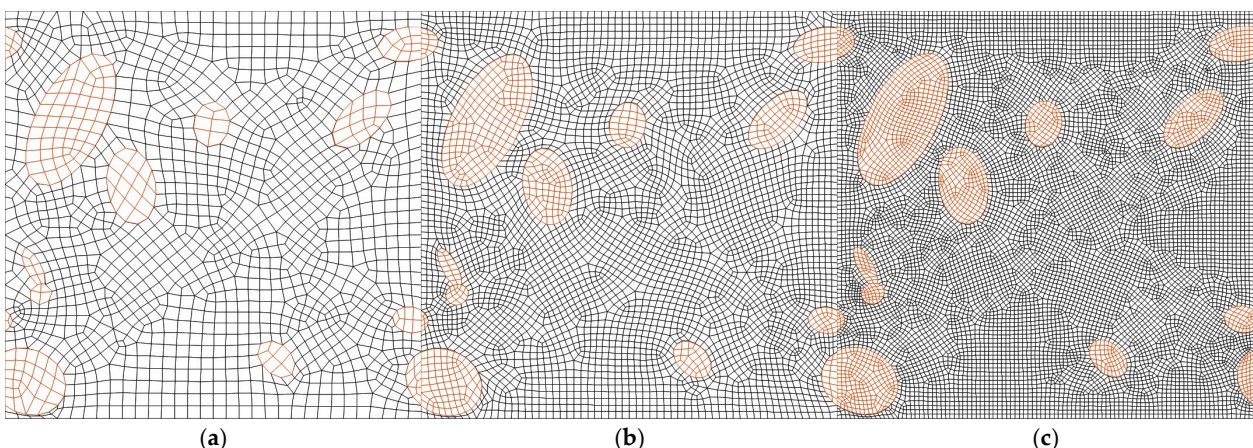

**Figure 7.** Example of meshes used for the medium-B RVE model with the following: (**a**) 262 elements/$\mu m^2$ and 1463 elements in the RVE, (**b**) 800 elements/$\mu m^2$ and 4477 elements, (**c**) 1881 elements/$\mu m^2$ and 10,519 elements. Si particles are highlighted in brown color.

**Table 2.** Density and number of brick elements within the 2.5D RVE.

| Model | Number of Brick Elements per $\mu m^2$ (Total Number of Brick Elements) | | | | | | | |
|---|---|---|---|---|---|---|---|---|
| Small 5 particles | | | | | | 1162 (3248) | 1991 (5566) | 2847 (7960) |
| Medium-A 10 particles | 178 (997) | 453 (2536) | 527 (2950) | 639 (3574) | 771 (4314) | 1299 (7266) | 1916 (10,718) | 2863 (16,014) |
| Medium-B 10 particles | 262 (1463) | | 539 (3017) | 800 (4477) | 975 (5452) | 1382 (7732) | 1881 (10,519) | 2835 (15,858) |
| Visualization | Figure 7a | | | Figure 7b | | | Figure 7c | |
| Large 15 particles | 173 (1448) | | | | | 1351 (11,334) | 1893 (15,879) | 2903 (24,347) |

In our study, the interfaces between elements are defined as continuous zones where a single displacement is associated with the nodes at the interface. It ensures that the mechanical response is consistent across the interface, facilitating stress and strain distribution throughout the model. This continuous interface allows for effective load transfer between adjacent elements, which is crucial for simulating realistic material behavior until decohesion happens. To study fracture and damage evolution, a future extension of the FEM simulations will serve to replace this continuous interface by cohesive elements.

A full 3D RVE is also built and meshed, with a cube matrix and ellipsoid particles. The mesh consists in this case of second degree tetrahedron elements, as automatic unstructured hexahedral meshes are not available. For the generation of a 3D RVE from 2D measured data, it is necessary to make some assumptions. Let us remember that any Si particle requires 9 numbers in 3D to be defined: the 3 coordinates of the centroid, the dimensions of the 3 axes, and the 3 Euler angles describing its orientation. It is necessary to optimize the 3D particle features to define a representative 3D RVE from the 2D experimental microscopic images. The optimization methods most commonly used are gradient methods but they are highly dependent on the initial coordinates and are subject to becoming trapped in local minimum. The gradient computation is resource consuming and binds the variables' changes together by defining a direction of evolution. When the number of variables to be optimized is important, an algorithm from the metaheuristics family is preferred because it allows solving complex problems without requiring the gradient computation. In this study, the optimization problem will be solved by using a genetic algorithm.

A genetic algorithm relies on several steps regardless of the problem to solve. Considering the case of a 3D RVE containing 10 ellipsoidal Si particles, 90 variables are to be

optimized. The first step consists of producing an initial population composed of several hundreds of characters, each one representing the properties to describe 10 ellipsoids. The next step consists of computing a fitness criterion (i.e., the optimization criterion) for each character. In this case, the "fitness criterion" allows describing the ability of the ellipsoid properties to model a representative microstructure of the 2D microscopic images. The choice of the optimization criteria will be described in the following paragraph. If the current set of ellipsoids reaches the convergence criterion, the algorithm halts. Otherwise, the characters are ranked based on their fitness, with the top half chosen as parents to produce the next generation. Each parent transmits 50% of its genes to a new character (i.e., 50% of the ellipsoid properties). Mutations can then occur and can affect the properties of the ellipsoids. The mutations are added to the genetic algorithms in order to limit the convergence to local minima. The first 10 percent of the ranked characters are retained as is. These operations are then repeated until the algorithm converges.

To simplify the optimization problem, two successive genetic algorithms were used. The first one allows optimizing the ellipsoid dimensions and their orientation. The dimensions of the ellipsoids were generated from a uniform distribution within a range established from the histograms presented in the Supplementary Materials. The Euler angles were also generated from a uniform distribution. For the computation of the first optimization criterion, the intersections of the ellipsoids have been computed for several tens of planes defined by random normal vectors. This allows estimating 2D statistics which can then be compared to the measured statistics. In order to facilitate the implementation of the genetic algorithm, it is preferable to keep a single optimization criterion. The first criterion corresponds to the sum of squares of the frequency differences between the experimental histograms and those obtained from the intersections. The orientation histogram is not included in this calculation of the criterion as orientation is a periodic variable and the considered interval depends on a reference frame that would be arbitrary in the case of random normal vectors to define cutting planes. The histogram is afterwards generated to check the absence of preferential orientation.

Once the first algorithm converges, a total volume can be determined for the best set of properties that allows us to calculate the dimensions of a cubic RVE for a given fraction. The second genetic algorithm then optimizes the centroid coordinates to limit the intersections. These coordinates were generated from a uniform distribution between 0 and the RVE side length. In this algorithm, the optimization criterion is the sum of the intersection volumes between each pair of ellipsoids, considering the periodicity of the RVE. Four 3D RVE models are defined by this method with 5, 10, 15, and 20 Si particles. The particles of the largest model are described in Table S2 in the Supplementary Material, while Table 3 provides the mesh density of the 3D RVE models chosen to be related to the ones used in 2D. Note that the 3D RVE FEM simulations are conducted using the METAFOR code v3494 [36], with second degree Tetrahedron elements preventing volume locking and implemented accordingly [40].

**Table 3.** Density and number of brick elements in the 3D RVE.

| Model | Number of Brick Elements per $\mu m^2$ (Total Number of Brick Elements) | | |
|---|---|---|---|
| 3D-5P 5 particles | 1124 (29,337) | 1857 (62,290) | |
| 3D-10P 10 particles | 1129 (73,042) | 1864 (155,027) | |
| 3D-15P 15 particles | 1213 (95,570) | 1681 (155,926) | |
| 3D-20P 20 particles | 979 (109,816) | 1153 (140,224) | 1597 (228,729) |

Within all RVEs, isotropic Hooke's law is adopted to model the elastic behavior of Si particles, and due to their very high yield limit (about 7000 MPa), no plastic behavior is assumed. Regarding the $\alpha$-Al matrix, an isotropic elasto-plastic (J2) model is applied with an isotropic hardening law described by the Voce law in Equation (3), as used in other research focused on Al alloys [41–43]:

$$\sigma_F = \sigma_0 + K\left(1 - exp(-n.\varepsilon^{pl})\right) \tag{3}$$

where $\sigma_F$ is the updated yield stress, $\varepsilon^{pl}$ is the plastic strain, and $\sigma_0$, $K$, and $n$ are the material data.

### 3.2. Boundary Conditions and Loading

The 2D periodicity of the boundary conditions is ensured by imposing relationships between the displacements of the nodes along all the edges of the FEM model, as shown in Figure 8a. The equations linking the displacements in the XY plane of the edge nodes are detailed in Table 4 for the 2.5D case. They ensure that the sides *b* and *d* keep the same deformed shape as the sides *a* and *c*, respectively, throughout the whole tensile process. An external node, called RY, and linked to the nodes along the sides *c* and *d*, is used to load the model in the Y direction. Extensions of these classical periodic conditions are applied to each face of the 3D RVE.

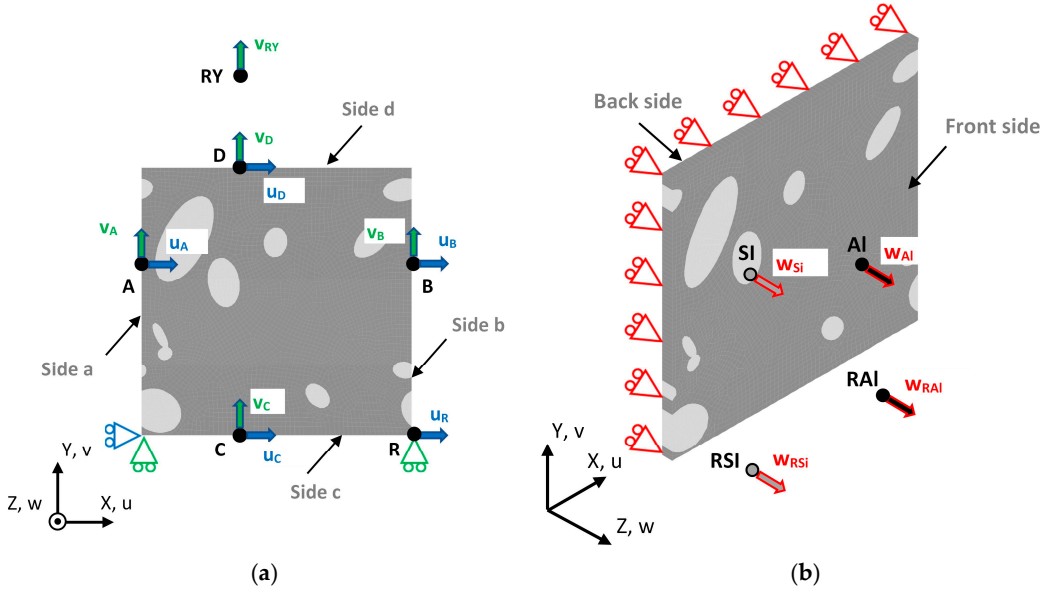

(a)          (b)

**Figure 8.** Boundary conditions of 2.5D RVE model: (**a**) fixation and displacements in XY plane, (**b**) fixation and displacements in Z direction, notation: edge nodes A, B, C, D, sides a, b, c, d.

**Table 4.** Description of the links between the displacements (u, v) of nodes A, B, C, D along the edges a, b, c, d in X and Y directions.

| Constraints in X Direction | in Y Direction |
|:---:|:---:|
| uB = uA + uR | vB = vA |
| uD = uC | vD = vC + vRY |

However, for a 2.5D RVE, the boundary conditions in the XY plane are not sufficient. The displacement conditions in the Z direction must also be adjusted to achieve a three-dimensional stress state similar to that of a 3D model. When simulating a simple tensile test along the Y direction with an RVE model without conditions on the third direction Z and the out-of-plane stresses, it was observed that the lateral strains, $\varepsilon_{XX}$, are almost

uniform in each Si particle and are also very close from one particle to another. Moreover, since the simplified 2.5D model should behave similarly to a 3D RVE model for this uniaxial tensile test, both the lateral strains $\varepsilon_{XX}$ and $\varepsilon_{ZZ}$ should be almost homogeneous in each Si particle. Globally, for each finite element, the lateral strains along the X and Z directions should be close.

Within the 2.5D RVE simulations, two external nodes (*RAl*, *RSl*) (Figure 8b) are used to control the out-of-plane strain state with linear elastic relations between the displacements along the Z of the nodes of the front side ($w_{Al}, w_{Si}$) and of the external nodes ($w_{RAl}, w_{RSi}$). The spring-like finite elements expressing these elastic relations (hereafter referred to as "springs") are calibrated by their stiffness values ($k_{Al}, k_{Si}$), for the elements belonging to the α-Al matrix and to the Si particles, respectively.

Since these stiffness values ($k_{Al}, k_{Si}$) depend on the nodal densities in each phase and must be defined for each model, two macro stiffness parameters, $K_{Al}$ and $K_{Si}$, common to all models and representing the stiffness of all the springs per unit area, have been defined in Equations (4) and (5). These last macro-data are consequently independent from the model size and from the mesh density. They were used to define the specific values of $k_{Al}$ and $k_{Si}$ in any 2.5D RVE model.

$$K_{Al} = \frac{n_{SA} \, k_{Al}}{dimx \, dimy \, P_{Al}} \tag{4}$$

$$K_{Si} = \frac{n_{SS} \, k_{Si}}{dimx \, dimy \, P_{Si}} \tag{5}$$

where $n_{SA}$ and $n_{SS}$ are the total number of springs in the Z direction attached to the nodes of the Al and Si elements, *dimx* and *dimy* are the dimensions of the 2.5D RVE model in the X and Y directions, and $P_{Al}$ and $P_{Si}$ are the surface ratio covered by the matrix and particle, respectively. The interface nodes attached to both the matrix and particles are considered attached to the Si material.

## 4. Law Identification of 2.5D RVE Model and 2D RVE Versus 2.5D RVE Results

The constitutive laws used for each phase, the material dataset, and the interface behavior between the Al and Si phases are key ingredients for the quality of the RVE results. In previous studies [33,34], the Young modulus of particles and the material behavior for each phase were separately measured by nanoindentation tests. Both Poisson ratios were estimated to be equal to 0.3, consistent with the values found in the literature. The numerical simulation of a macro tensile test with the medium-B RVE model and a fine mesh (model shown in Figure 7c with 1881 elements/μm$^2$ and a total of 10,519 brick elements) is used to adjust the Voce material parameter and the numerical parameters $k_{Al}, k_{Si}$ of the model by the inverse method.

More specifically, the data of the constitutive law of the matrix ($\sigma_0$, *K*, and *n* in Table 5) are identified by minimizing the difference between the numerical prediction and the average measured macro tensile curve (see curve "Exp-aver" in Figure 9a). These parameters, close to those determined from the indentation measurements [33,34], allow recovering the experimental stress–strain curve (see Figure 9a). The spring stiffnesses values ($k_{Al}, k_{Si}$) of the 2.5D model are adjusted by reducing the deviation between the deformations $\varepsilon_{XX}$ and $\varepsilon_{ZZ}$ over all the elements of the model (Table 6). This result is used to compute the macro stiffnesses values $K_{Al}$ and $K_{Si}$, through Equations (4) and (5). Those generic values are exploited to compute the spring stiffnesses of all the other models (small, medium, large) and all the mesh densities. Note that the applied α-Al matrix behavior identification methodology required less experimental data than the former approach based on indentation experiments.

**Table 5.** Material data of the Si particles and the $\alpha$-Al matrix defined for the 2.5D RVE model.

| Si—Elastic Law | | $\alpha$-Al—Elastic Law | | $\alpha$-Al—Voce Law | | |
|---|---|---|---|---|---|---|
| *E* [MPa] | $\nu$ | *E* [MPa] | $\nu$ | *K* [MPa] | $\sigma_0$ [MPa] | *n* |
| 167,000 | 0.3 | 83,744 | 0.3 | 163.0 | 176.7 | 21 |

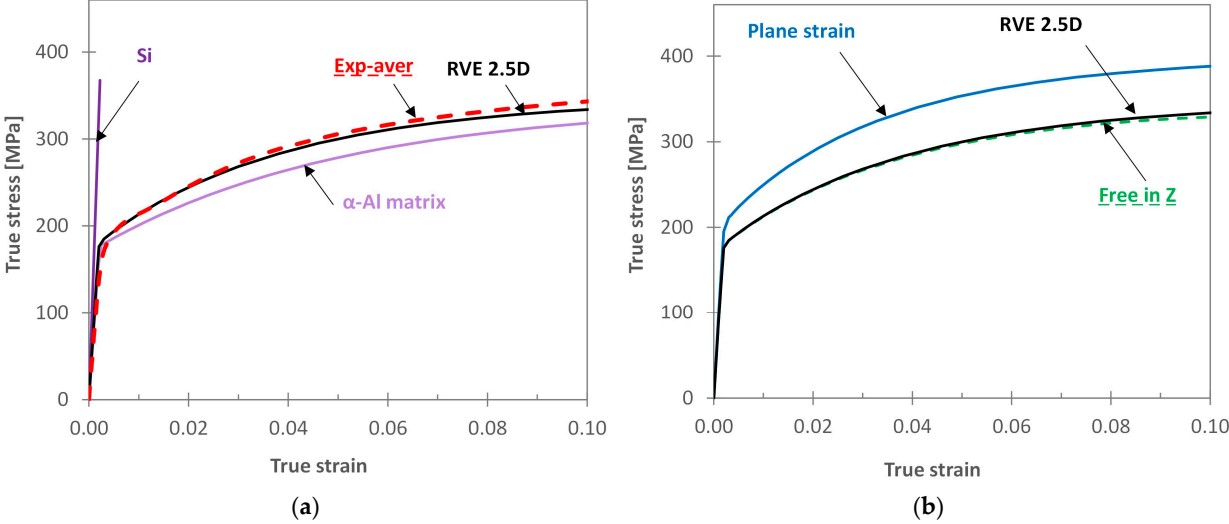

(a)  (b)

**Figure 9.** (**a**) Comparison of the behavior of the $\alpha$-Al matrix, one Si-rich particle, the macroscopic behavior predicted by the 2.5D RVE model, and the average experimental tensile curve (Exp-aver), (**b**) comparison of the stress–strain curves for the plane strain state, the "Free in Z", and the 2.5D RVEs.

**Table 6.** Spring stiffness values for out-of-plane displacements with $k_{\text{Al}}$ and $k_{\text{Si}}$ identified for the medium-B RVE model and the fine mesh shown in Figure 7c and $K_{\text{Al}}$ and $K_{\text{Si}}$ used in any model (Equations (4) and (5)).

| Stiffness per Spring [N/mm] | | Stiffness per Area [N/mm$^3$] | |
|---|---|---|---|
| $k_{\text{Al}}$ | $k_{\text{Si}}$ | $K_{\text{Al}}$ | $K_{\text{Si}}$ |
| 0.0121 | 2.11 | $2.27 \times 10^7$ | $4.66 \times 10^9$ |

Figure 9a also shows the behavior of the two phases present in the composite material ($\alpha$-Al matrix and Si-rich particles). The 2.5D RVE developed in this research is compared in Figure 9b with two 2D RVEs: a classical membrane one, in the plane strain state, and a 2D RVE called "Free in Z" where the out-of-plane springs are disabled, which behaves like a thin membrane composed of 3D elements. These results show the effect of the out-of-plane boundary conditions on the predicted stress–strain curves. The plane strain state is far too stiff to correctly predict the experimental behavior of the material, while the 2.5D RVE and 2D Free in Z predictions and experiments overlap. Although the difference between the 2.5D and "Free in Z" for macro predictions seems negligible, the local distribution of stress still needs to be analyzed. Looking at the AA cross-section passing through several Si particles and passing through a zone where the deformations are high (Figure 10a), one can easily verify that the distributions of strains and stresses are indeed close in the X and in the Z directions in the 2.5D RVE model (Figure 10b,c) for Si particles and for the $\alpha$-Al matrix. The predictions the local stress components XX and ZZ, $\sigma_{XX}$ and $\sigma_{ZZ}$ respectively, along cut AA of the two models, "Free in Z" and 2.5D RVEs, are further investigated in Figures 11 and 12. These plots underline the effect of the specific boundary conditions on these stress components. The stress distribution of the 2.5D RVE model is closer to the expected physical isotropic behavior.

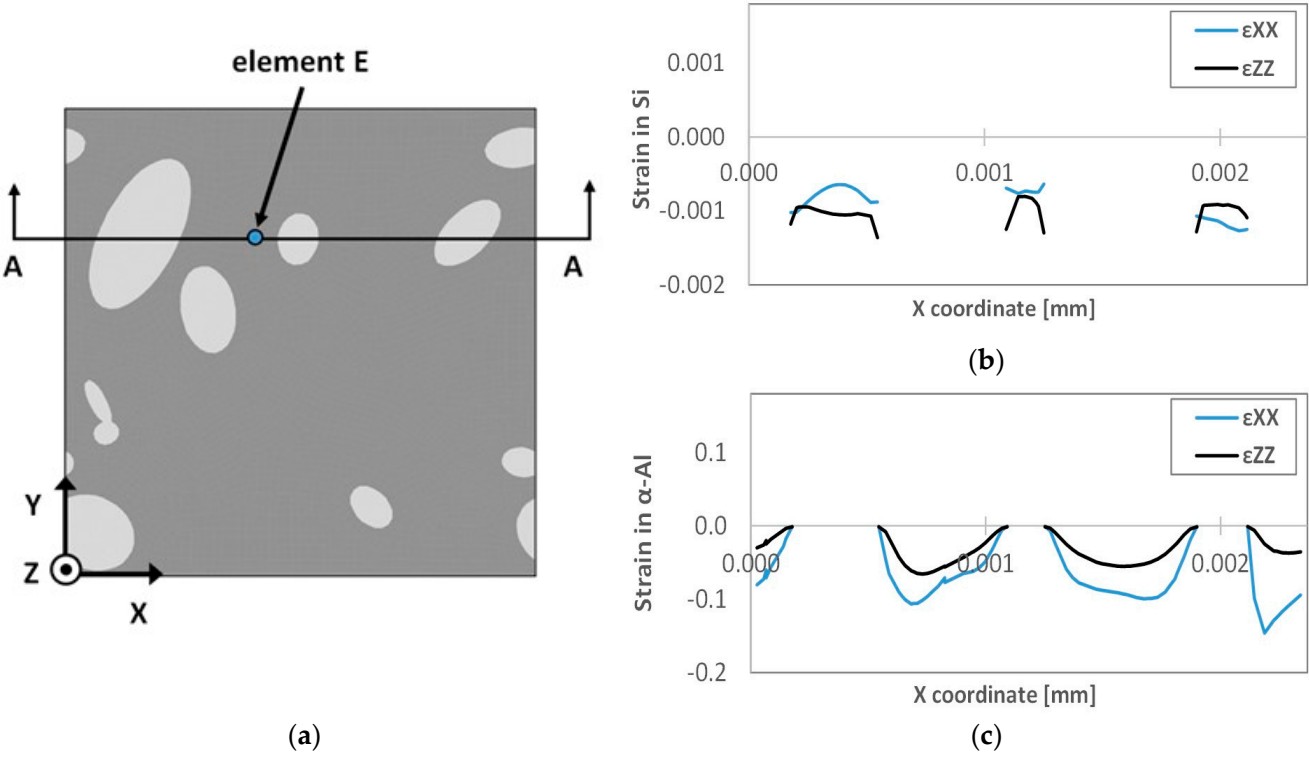

**Figure 10.** (**a**) Position of section AA and of the element E in the 2.5D RVE model, (**b**,**c**) strain distribution along cut AA in the 2.5D RVE for a macro strain of 10% in Y direction, (**b**) in Si particles, (**c**) in α-Al matrix.

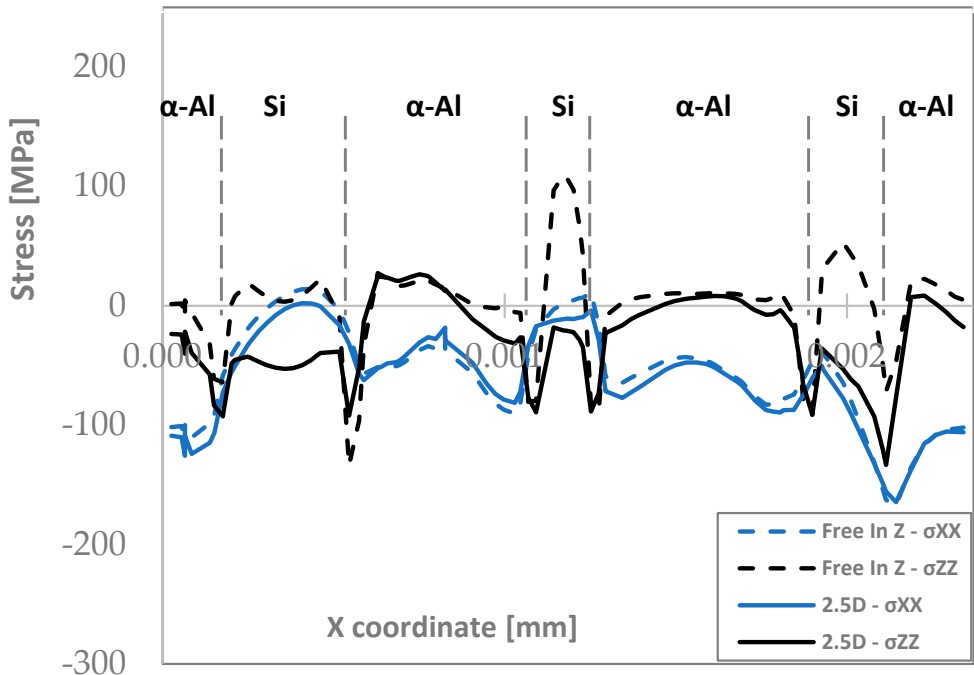

**Figure 11.** Comparison of the stress distribution along cut AA for a macro strain of 10% in the Y direction: in the "Free in Z" model and in the 2.5D RVE model.

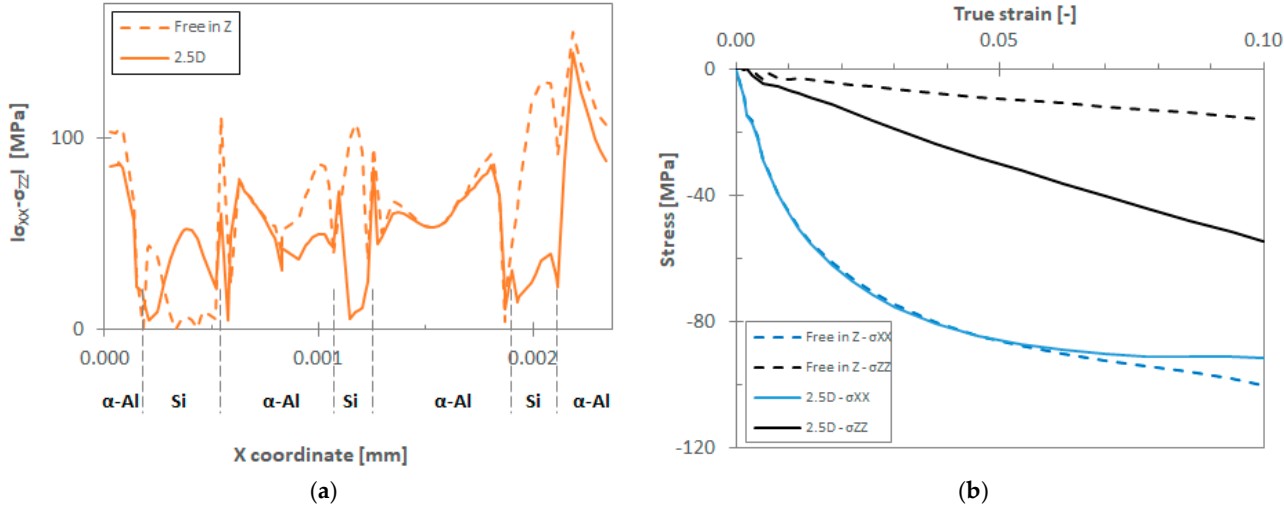

**Figure 12.** Comparison of the stress state in 2D RVE "Free in Z" and 2.5D models for a macro strain of up to 10% in the Y direction: (**a**) absolute difference between $\sigma_{XX}$ and $\sigma_{ZZ}$ along cut AA direction, (**b**) stress evolution in the element E.

The decrease in the difference between $\sigma_{XX}$ and $\sigma_{ZZ}$ in the 2.5D RVE model compared to the 2D RVE "free in Z" case is further quantified in absolute value in Figure 12a for cut AA, while the stress evolution in the E element during the tensile simulation (see element E position in Figure 10a) confirms the improvement of the "isotropic" behavior in an average sense in Figure 12b for the 2.5D RVE compared to the "Free in Z" RVE model. Indeed, due to the material macroscopic isotropy, the Y tensile load should not affect the Z and X stress and strain fields differently on average. To quantify the improvement of the stress state in the 2.5D RVE model, Equation (6) computes the average difference $\Delta$ between the two stress components $\sigma_{XX,i}$ and $\sigma_{ZZ,i}$, for a macro strain of 10% in the Y direction, considering all the elements along cut AA and throughout the entire model. Three models are compared: the membrane in the plane strain state, the "Free in Z" model, and the 2.5D RVE. All the values pertain to the medium-B model and use the same fine mesh. Table 7 shows that the 2.5D model better describes the behavior of the material thanks to the decrease in the $\Delta$ value.

$$\Delta = \frac{\sum_{i=1}^{n} |\sigma_{XX,i} - \sigma_{ZZ,i}|}{n} \tag{6}$$

**Table 7.** Comparison of the average difference $\Delta$ [MPa] along cut AA and in all the elements of the models for a macro deformation of 10% in the Y direction, in the plane strain state, the "Free in Z", and the 2.5D RVEs.

| Cut AA | | | Whole Model | | |
|---|---|---|---|---|---|
| **Plane Strain** | **Free in Z** | **2.5D** | **Plane Strain** | **Free in Z** | **2.5D** |
| 181 | 69.4 | 52.5 | 182 | 36.2 | 31.8 |

## 5. Result Analysis and 2D/3D Validation

All the 2.5D RVE models (5, 10, and 15 elliptical Si-rich particles with different particle distributions, different mesh sizes, and node densities) and 3D RVE models (5, 10, 15, and 20 elliptical Si-rich particles and different mesh sizes) presented in Tables 2 and 3 are used for a sensitivity and convergence analysis. A single input dataset (out-of-plane macro stiffness parameters $K_{Al}$ and $K_{Si}$ (Table 6) and material data of Table 5) is applied within all these simulations. A uniaxial tensile test up to a macro strain of 10% is modeled, and the FEM simulation results of a "20 particles_3D RVE refined mesh" as well as a

"10 particles_2.5D RVE Medium B mesh" are compared, with the experiment in Figure 13a showing good accuracy.

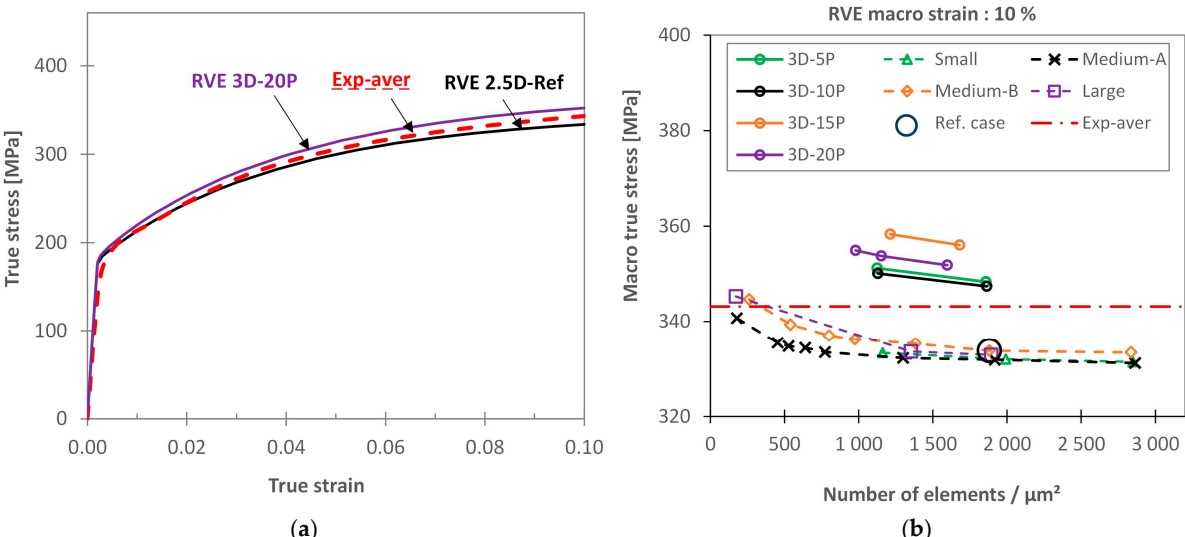

**Figure 13.** Comparison of the experimental macro stress (Exp-aver) with the predictions of 2.5D RVE and 3D RVE models: (**a**) tensile test up to 10% in the Y direction, (**b**) the final macro stress for a macro deformation of 10%, for all the 2.5D RVE and 3D RVE models of Tables 2 and 3.

The macro-(true) stresses, calculated as the applied force associated with an imposed displacement corresponding to a true strain of 10%, divided by the average cross-sectional area relative to each RVE case and mesh density, are compared in Figure 13b. From a macroscopic point of view, all the 2.5D RVE models (small, medium-A, medium-B, large) are quite similar. They converge to a single value, slightly underestimating the experimental average stress, while the scattering between the 3D RVE results is a little larger, and the 3D RVE results present an overestimation of the experiment.

For the 2.5D RVE and microstructure studied, a fine mesh with around 1900 elements per $\mu m^2$ can accurately predict the macro stress behavior. A logic curve convergence based on mesh densities and RVE sizes is observed, and the impact of particle distribution is quantified between medium-A and -B simulations, confirming that 10 particles is enough in 2D RVE simulations to have a reliable answer. Note that the medium-B model with 10 particles, 1881 elements per $\mu m^2$, and named "Ref. case" in Figure 13 (hollow circle dot) was the one used to identify the behavior of the $\alpha$-Al matrix material (Table 5) and to analyze the local stress and strain fields.

In 3D simulations, a complete convergence analysis on mesh densities and RVE sizes was not performed; however, refining the mesh decreases the macro stress computed at a 10% strain and increases the accuracy. This indicates that a coarse mesh cannot adequately handle the high strain and stress gradients near the matrix–particle interface due to the material stiffness differences.

One might argue that the relative position of the results for the different number of particles (also defining the RVE size) in 2D or 3D RVEs, as shown in Figure 13b, appears erratic beyond the positive effect of mesh refinement. However, the authors believe that this just confirms that all these RVE sizes are reliable for predicting the macroscopic curve, provided the mesh refinement is correct (see Figure 14). The results also show a difference between a 2.5D RVE and 3D RVE, justifying the effort to consider boundary conditions in the transverse direction. Figure 14 indicates that using 5 to 20 particles in a 3D RVE still results in some scattering like the physical experiments. While larger RVEs would reduce this scattering, this investigation was not performed as the main goal of the article is to save computational costs and develop a 2.5D RVE approach.

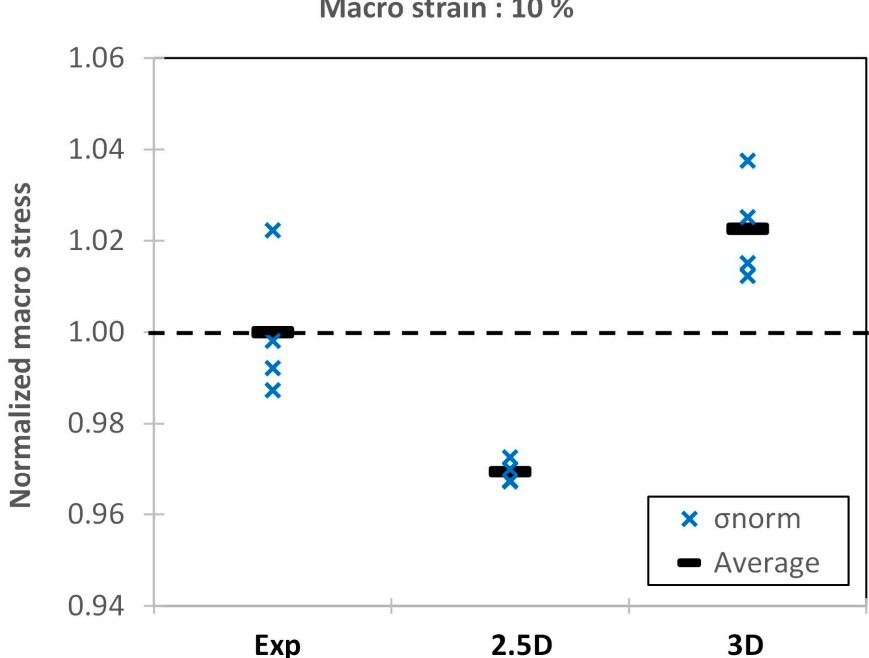

**Figure 14.** Normalized macro stresses and their average values for the experiments (Exp) and the 2.5D RVE (5, 10A, 10B, 15 particles) and 3D RVE (5, 10, 15, 20 particles) simulations for a mesh density around 1800 elements per μm².

Figure 14 shows that the average responses of 2.5D RVE and 3D RVE predictions fall within or close to the scattering observed in experimental tests (4%). Even though the 3D RVE demonstrates a better accuracy, the 2.5D RVE maintains an error of less than 4% and remains within the standard experimental deviation.

The result scattering between the 2.5D and 3D RVE models is due to the high sensitivity of the maximum local stress to the relative position of the particles. The small model with five particles predicts a lower maximum stress due to its low number of particles unable to represent all the possible interactions between the particles and matrix as in the real material. On the contrary, when the number of particles increases up to 10, like in the medium-A, the medium-B, or for the 15 particle models in the 2.5 D RVE, the gap between the maximum stresses in the simulations is reduced. The same observation is also found for the maximum stresses in the transverse and in the out-of-plane directions or in the 3D RVE models.

Experimental material isotropy is validated in the 2.5D RVE for 10 particles and for 3D RVE for 15 and 20 particle cases. A minimum number of particles is indeed required to correctly model the material behavior. For the 2.5D RVE, Figures 15 and 16 plot the strain and the stress fields, respectively, according to the directions X (a) and Z (b) at the end of the simulation for a macro strain of 10% and for the "Ref. case". These figures confirm that similar stress and strain states are observed in both the X and Z directions and that, of course for a Y tensile direction, the internal transversal stress is very low only due to the differences in matrix and particle strengths. The internal total transversal strains have to account for the elastic volume changes and the plastic deformation heterogeneity due to the particles. The equal local strain fields in the X and Z directions in any cut of the 2.5D RVE are far from being exactly reached, as the macroscopic equality is applied as a light constraint, which just helps to define a consistent behavior in the Z direction within the 2.5D RVE simulation. Moreover, Figures 15 and 16 confirm the efficient implementation of the periodic boundary conditions.

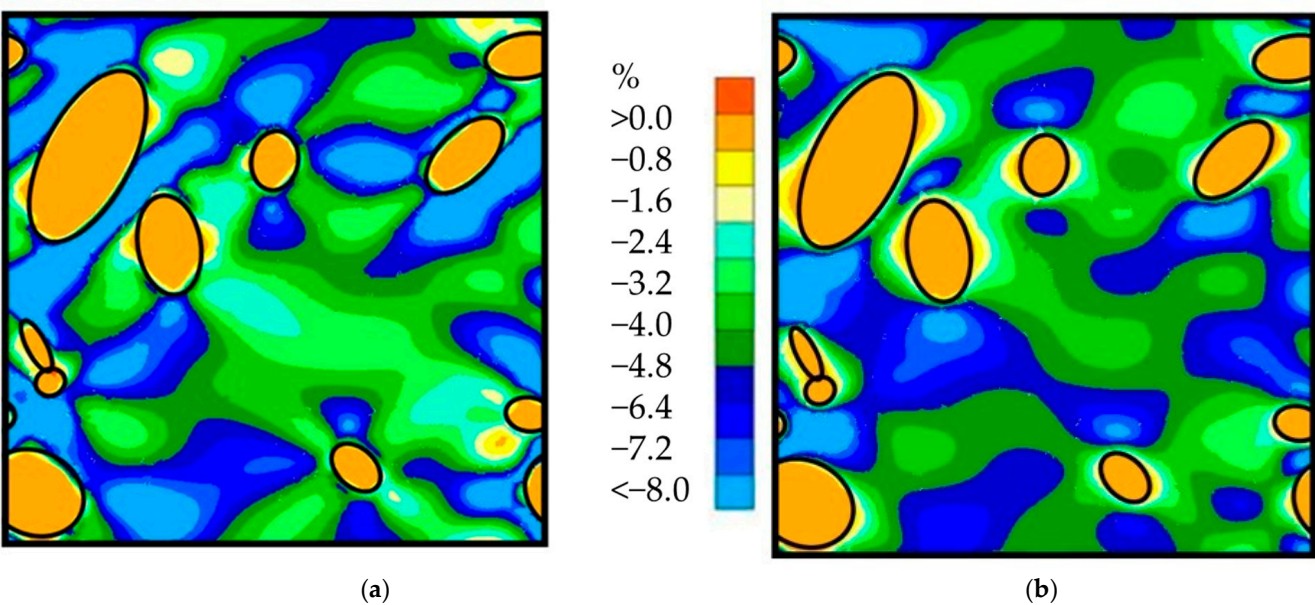

**Figure 15.** Local strain fields predicted by the 2.5D RVE Ref. model for a macro tensile strain of 10% in the Y direction: (**a**) $\varepsilon_{XX}$, (**b**) $\varepsilon_{ZZ}$. Si particle contours are highlighted in black.

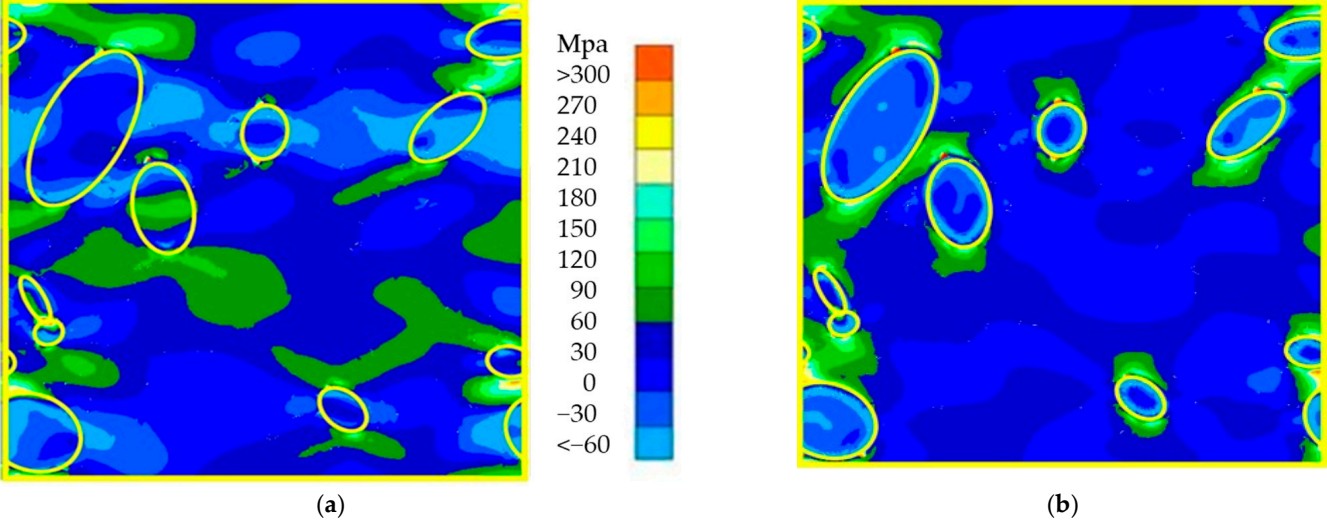

**Figure 16.** Local stress fields predicted by the 2.5D RVE Ref. model for a macro tensile strain of 10% in the Y direction: (**a**) $\sigma_{XX}$, (**b**) $\sigma_{ZZ}$. Si particle contours are highlighted in yellow.

Figure 17 presents the 2.5D RVE distribution of strains (a) and stresses (b) in the loading direction (Y). The critical areas are at the matrix–particle interface with risks of decohesion and where the experiment predicts rupture initiation [15]. One could even see in Figure 17a the beginning of a strain localization between some particles. The current model without cohesive elements at a matrix–particle interface does not include the damage evolution present at the interface in the real material for a macro strain larger than 0.10. The degradation of the interface is probably already beginning earlier, so no quantitative interpretation was completed here within the current results, assuming continuous interfaces. However, the predicted trends seem consistent with the damage mechanisms already experimentally identified in [15] (decohesion, localization, rupture of particles starting around 10% strain).

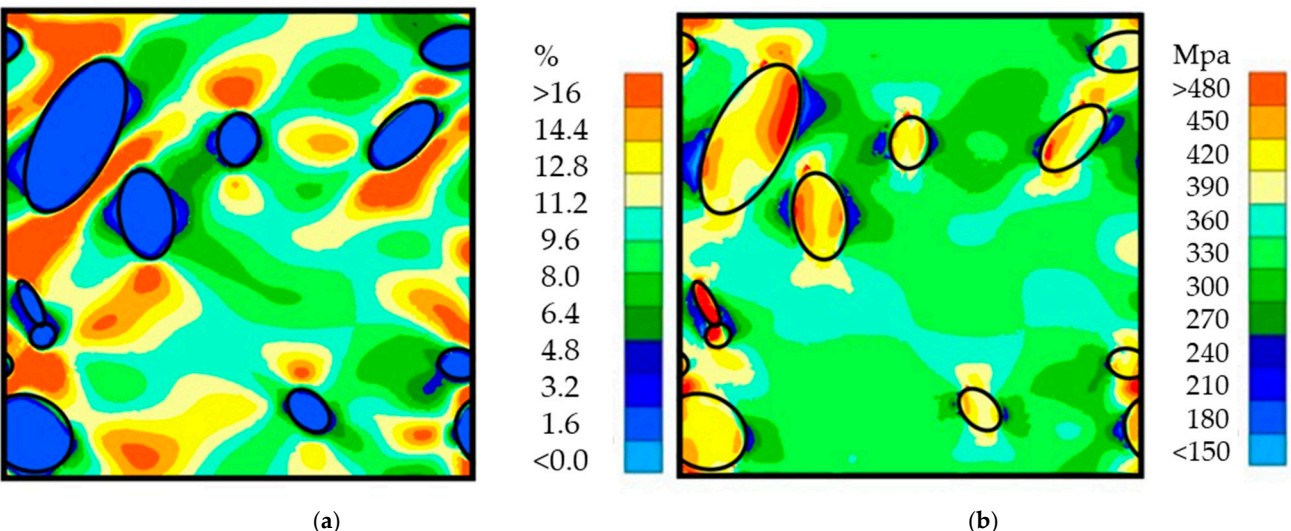

**Figure 17.** Strain and stress fields predicted by the 2.5D Ref. RVE model for a macro tensile deformation of 10% in the Y direction: (**a**) $\varepsilon_{YY}$, (**b**) $\sigma_{YY}$. Si particle contours are highlighted in black.

The local stress fields (Figures 15–19) computed by the Lagamine (2.5D RVE) and METAFOR v 3494 (3D RVE) codes show a close agreement for the matrix stress value, even if the distribution of particles differs between the 2.5D square and the 3D cube, and if these stresses are computed by different element types. These RVEs are built for the same particle statistics (see Image J post-processing in Section 2), so indeed both models should provide similar results on average.

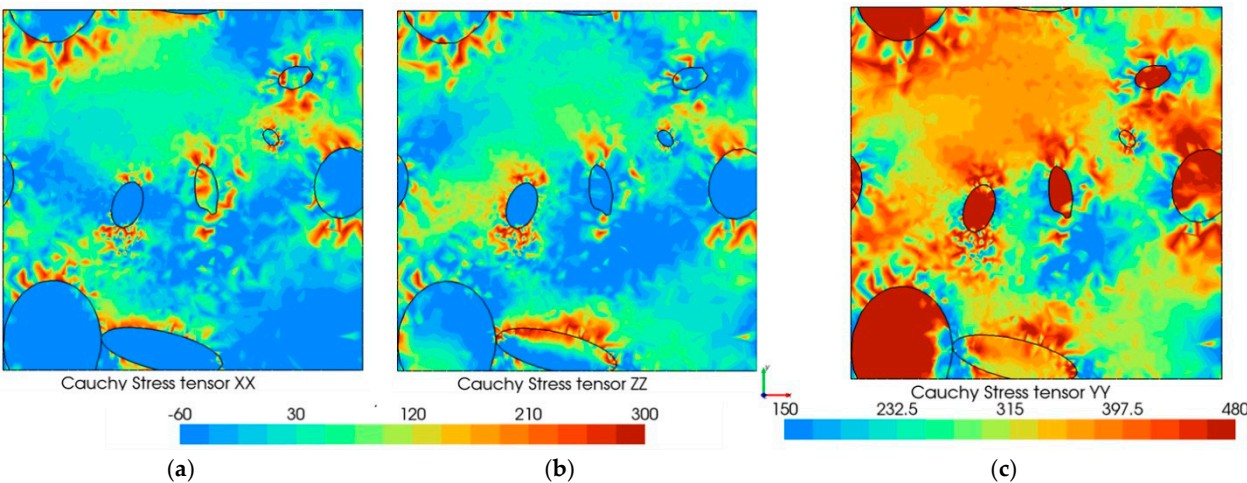

**Figure 18.** For a macro tensile deformation of 10% in the Y direction, (**a**) predicted stress $\sigma_{XX}$ and (**b**) $\sigma_{ZZ}$ by 3D RVE 20 particles (METAFOR v 3494 software) in (MPa) with a scale similar to Figure 16 (Lagamine software, 2.5D RVE) and stress $\sigma_{YY}$ (**c**) with a scale similar to Figure 17b.

A direct comparison of the local fields between 2.5D and 3D RVE simulations is not straightforward, as even if built for the same particle statistics, the particle distribution methodology used does not impose a similar choice of particles. For the 3D RVE (Figure 18) and 2.5D RVE (Figures 16 and 17), simulations with a similar mesh density and edge size are presented. The computed local stress fields are in close range. In agreement with the predicted global macro stress shown in Figure 14, the 2.5D RVE predicts local stresses as being slightly lower than the 3D RVE results. The level of the 2.5D RVE results is sensitive to the transversal boundary condition optimized to recover the macroscopic stress–strain curve. As shown in Figure 9, the plane strain is definitively too stiff and the Free Z condition

does not allow having a similar stress state in the X and Z directions (Figure 12), as expected and indeed predicted by the 3D RVE simulations (Figure 18a,b).

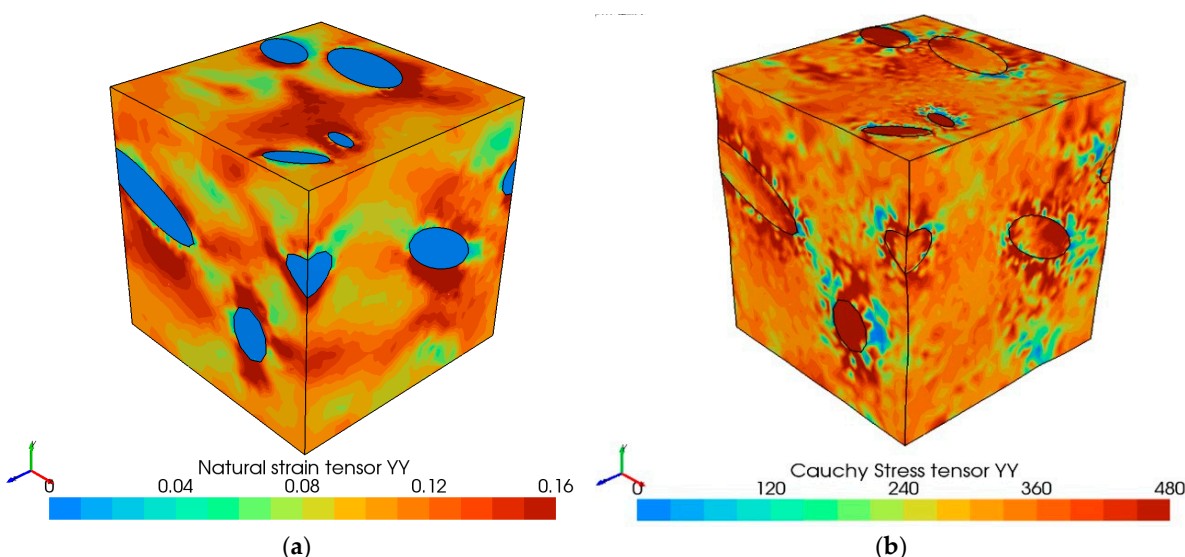

**Figure 19.** For a macro tensile deformation of 10% in the Y direction, predicted strain $\varepsilon_{YY}$, (**a**) and stress $\sigma_{YY}$ in (MPa) (**b**) by a 3D RVE 20 particles (METAFOR v 3494 software).

The 3D RVE simulation result in the Y direction (Figure 19) confirms the strong heterogeneity of the stress and strain fields close to the particles. The 2.5D RVE model presents, versus a 3D RVE one, a high CPU advantage as the associated simulation times are decreased by a factor 417 (real time) or 2581 (CPU time). The parallel computing distribution is only interesting for the 3D RVE. This computing time comparison was performed on an AMD Thread Ripper 3970X 32-Core Processor (PRIMINFO, Liège, Belgium), exploiting16 threads for similar mesh densities.

## 6. Conclusions

A methodology for constructing a 2.5D RVE that improves the FEM prediction of local stress and strain fields in an isotropic two-phase material (matrix and particles) has been described. Less accurate than a 3D RVE simulation but improved compared to classical 2D RVE approaches, a 2.5D RVE provides a quick alternative for the identification of particle shapes and sizes, generating an optimal tensile behavior.

The tensile behavior of the L-PBF AlSi10Mg material post-processed by FSP and presenting a soft matrix compared to stiff particles has been predicted by RVEs. The high local gradients near the matrix–particle interfaces, which corresponds to the experimentally observed damage modes, were computed. The simulation results show that all RVE simulations could predict stress–strain curves. The plane strain 2D RVE evaluates a tensile curve with an excessive stiffness compared to the macroscopic experiment. The "Free in Z" 2D RVE and 2.5D RVE predictions are quantitatively close to the experimental values. By analyzing the strain fields according to the X and Z directions for a tensile test in the Y direction, the isotropic behavior is better recovered in an average way for a 2.5D RVE than for a "Free in Z" 2D RVE assumption. Cohesive elements would be necessary to quantitatively simulate static failure (strain above 10% not investigated here).

The key advantage of the 2.5D model is its computational efficiency. The identified microstructure by a 2.5D RVE could speed up an accurate 3D RVE optimization. Indeed, the CPU time of a 3D RVE becomes an issue if optimization loops, deep learning training, or FE$^2$ computation are foreseen. A long-term goal could be the extension of the 2.5D RVE with matrix–particle decohesion and advanced cyclic damage constitutive law to address the prediction of Wöhler curves. It should speed up material design for improved behavior in fatigue.

While the 2.5D RVE approach offers substantial computational efficiency and maintains accuracy in many cases, there are situations where the 3D RVE may be more suitable. For example, when dealing with materials that exhibit strong anisotropy or complex 3D microstructures—such as intricate grain orientations, strong anisotropy behavior of each particle, void distribution with a non-representative planar pattern, or fiber reinforcements—the 2.5D approximation will not fully capture the material behavior. In those cases, where out-of-plane stresses present strong heterogeneity between particles or where full 3D representation is necessary to characterize the microstructure geometry, the prediction of mechanical responses even under a uniaxial tensile or compression load with a 2.5D RVE will be inaccurate. Therefore, for materials with complex 3D microstructures, the 3D RVE remains the preferred choice to ensure accuracy across all spatial dimensions.

The uniaxial target loading is also a limitation, even for material presenting an isotropic behavior. Any macroscopic load where a relation between some average strain components can be found thanks to the material isotropic property could bring a methodology extension; however, complex loading will be excluded. While the current model successfully estimates the monotonic tensile curve, it may benefit from further extensions, such as incorporating kinematic hardening into the matrix constitutive model, to enhance its accuracy and applicability into cyclic scenarios.

In conclusion, the 2.5D method classifies different microstructures, speeds up material design, and saves resources compared with the 3D RVE method. The simplicity of the 2D RVE mesh operation as well as the short direct link from 2D SEM images to define a representative set of particles without complex tomography experiments or image reconstruction is of interest. Our qualitative comparison of a 2.5D RVE versus a 3D one has not pointed out a huge accuracy decrease in the local interface matrix–particle stress and strain field. So, future work should include a statistical comparison of 2.5D and 3D RVE results to identify if a factor relating 2.5D RVE local interface information to a 3D one for matrix–particle microstructure is required, as suggested by Qayyum et al. [9]. Cyclic loading simulations are also foreseen to address fatigue behavior. Purely numerical perspectives are numerous, such as an easy study of the effects of the proportions, sizes, shapes, orientations, and distributions of the particles, and of the strength ratio matrix/particle on mechanical properties, as long as the macroscopic behavior stays isotropic.

**Supplementary Materials:** The following supporting information can be downloaded at: https://www.mdpi.com/article/10.3390/met14111244/s1, Figure S1: Si precipitates: (a) cumulative occurrence based on their equivalent diameter $\varnothing_{eq}$, (b) distribution of particles with respect to their size. Figure S2: Si particle morphology and orientation: (a) aspect ratio, (b) major axis angle with respect to the X axis. Table S1: Description of the 10 elliptical Si particles in the medium-B RVE model. Table S2: Description of the 20 elliptical Si particles in the largest 3D RVE model (radii, volume, and Euler angles).

**Author Contributions:** Conceptualization, J.-P.P., A.M.H. and C.B.; methodology, L.D., C.B. and H.-S.T.; software, M.C., L.P. and C.B.; validation, M.C. and L.P.; formal analysis, F.C.; investigation, C.B.; data curation, H.-S.T.; writing—original draft preparation, C.B.; writing—review and editing, A.M.H., C.B. and F.C.; visualization, C.B. and L.P.; supervision, A.M.H. and L.D.; project administration, J.-P.P.; funding acquisition, J.-P.P. All authors have read and agreed to the published version of the manuscript.

**Funding:** This research was funded by Service Public de Wallonie Award 1810016 project LongLifeAM project of WALInnov program—Convention 1810016 Région Wallonne (Recipient J.P. Ponthot). As Research Director of F.R.S-FNRS, AM Habraken acknowledges the support of this institution (Fonds De La Recherche Scientifique—FNRS).

**Data Availability Statement:** Raw data as well as any RVE mesh can be shared as required.

**Acknowledgments:** We acknowledge also the fruitful exchanges with our colleagues C. van der Rest, J.G. Santos Macias from IMAP UC Louvain, and O. Dedry from ULiege about the microstructure of FSP LPBF AlSi10Mg.

**Conflicts of Interest:** The authors declare no conflicts of interest.

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
