# Peer review of "Efficient Representative Volume Element of a Matrix–Precipitate Microstructure—Application on AlSi10Mg Alloy"

_metals, doi:10.3390/met14111244_

Round 1

Reviewer 1 Report

Comments and Suggestions for Authors

Rapresentative Volume Elements (RVEs) are a way of approaching and discretizing the simulation of complex materials to be simulated such as composites and lattice structures. They represent a great opportunity with regard to simplifying and managing the simulation of various physical problems in a functional and targeted manner. The interest in models that are able to handle 3D elements by means of a simplification that approaches 2D models, and thus 2.5D, is a very interesting aspect in terms of the amount of resources that are used in the calculation and management of the results. The work is well structured and can be considered for publication in this form.

Reviewer 2 Report

Comments and Suggestions for Authors

The paper "Efficient Representative Volume Element of a matrix-precipitate microstructure –application on AlSi10Mg alloy" in a very detailed way describes the development of 2.5D RVE approach showing innovation and simultaneously practical application for this particular class of material simulation. It is agreed that the authors did manage to show the efficiency gain in computational time with high accuracy regarding its comparison with 3D RVE modeling. This work provides great insight, in particular, for material design optimization and a reduction in the required computational resources in applications involving additive manufacturing. Full validation of the 2.5D RVE model from experimental results increases the credibility of the approach. In general, the paper is well-structured and forms an important contribution to the area, while including both theoretical and practical developments.

It would be worthwhile to make a few additions/corrections:

1. Clarity of Description of Methodology: The article describes the 2.5D RVE approach in detail; nevertheless, the clarity could be substantially improved for readers unfamiliar with sophisticated simulation methodologies by adding further step-by-step illustrations and/or schemes.

2. Providing more explicit details regarding the experimental conditions used for validation, such as measurement uncertainties or potential sources of variability, could add robustness to the validation process.

3. The 2.5D vs. 3D RVE is well-justified regarding computational efficiency; however, some discussion of any possible drawbacks or 'edge cases' where the 2.5D approach may not work as well as 3D RVE would give balance to the discussion.

Reviewer 3 Report

Comments and Suggestions for Authors

The reviewed paper presents an interesting and useful work on the optimization of finite-element (FE) simulations for two-phase isotropic materials with taking into account statistics of the hard dispersed phase. To do this, the Authors consider a thin one-FE layer with virtual accounting of the material reaction in the reduced dimension by means of applied external spring elements ensuring approximate equality of the out-of-plane and in-plane strains in the directions perpendicular to the loading direction. The Authors term this approach as 2.5D RVE and compare it with constrained 2D RVE (plane strain) free in the out-of-plane direction 2D RVE and the full 3D model. Although difference in the macroscopic stress-strain curve between 2.5D RVE and free2D RVE is negligible, the developed approach provides isotropy of in-plane and out-of-plane directions, which is an essential property. The presented results can be used in further development of multi-scale material models. Thus, the paper can be recommended to publication in “Metals” after a minor revision.

1. Section 3.1: “The Euler angles were also generated from a uniform distribution in the interval 0 to 360 degrees”, this procedure will not provide a uniform distribution of orientations.

2. “The mesh consists in this case in linear tetrahedra elements as automatic unstructured hexahedral meshes are not available”, but “Note that the 3D RVE simulations are conducted using the METAFOR code [33] with Volume3DElement, a hexahedron with 8 nodes.” Which type of elements is used in 3D FEM?

3. Introduction: “conclusions and future perspectives are discussed in section 7” it must be section 6.

4. The abbreviation “DP” is not introduced at the first occurrence.

5. In Fig. 9, panel "(c)" is not indicated.

6. Although English is mostly fine, it must be additionally checked. For instance “…these local stress components within this cut are further investigated Fig.10… ”

Comments on the Quality of English Language

Although English is mostly fine, it must be additionally checked. For instance “…these local stress components within this cut are further investigated Fig.10… ”

Reviewer 4 Report

Comments and Suggestions for Authors

This work is interesting and belongs to journal scope however some clarification are required@

“ achieving the isotropy of the studied material”- how can be achieved isotropy for a material manufactured through AM that is highly anisotropic.

There was indicated “

 train a neural network” however before was no mentioned nothing about machine learning /optimization or AI the abstract is little bit unclear

how practically is to combine FSP with AM in the industrial scale ?

“Very low experimental result scattering is observed, and the average curve will be henceforth”  not clear how useful is to use an average for such scattering behaviour – rather a more large company for testing is required or further optimization should be provided. As no the results and approach taken  is out of work . This refer to figure 2 and 3

Not very clear how was defined the interface for element presented in figure 6

Eq is OK but this apply for this material ? I am little bit sceptical

Some recent literature is required

Comments on the Quality of English Language

Moderate editing of English language required.

Round 2

Reviewer 4 Report

Comments and Suggestions for Authors

.

Comments on the Quality of English Language

.